# Circular Patch Fed Rectangular Dielectric Resonator Antenna with High Gain and High Efficiency for Millimeter Wave 5G Small Cell Applications

**DOI:** 10.3390/s21082694

**Published:** 2021-04-11

**Authors:** Abinash Gaya, Mohd Haizal Jamaluddin, Irfan Ali, Ayman A. Althuwayb

**Affiliations:** 1Wireless Communication Centre, School of Electrical Engineering, Universiti Teknologi Malaysia, Skudai 81310, Malaysia; abinashgaya@gmail.com (A.G.); irfan_lrk_15@yahoo.com (I.A.); 2Electrical Engineering Department, Jouf University, Sakaka 72388, Aljouf, Saudi Arabia; aaalthuwayb@ju.edu.sa

**Keywords:** 5G, dielectric resonator antenna, aperture coupled, millimeter wave, 26 GHz, small cell

## Abstract

A novel method of feeding a dielectric resonator using a metallic circular patch antenna at millimeter wave frequency band is proposed here. A ceramic material based rectangular dielectric resonator antenna with permittivity 10 is placed over a rogers RT-Duroid based substrate with permittivity 2.2 and fed by a metallic circular patch via a cross slot aperture on the ground plane. The evolution study and analysis has been done using a rectangular slot and a cross slot aperture. The cross-slot aperture has enhanced the gain of the single element non-metallic dielectric resonator antenna from 6.38 dB from 8.04 dB. The Dielectric Resonator antenna (DRA) which is designed here has achieved gain of 8.04 dB with bandwidth 1.12 GHz (24.82–25.94 GHz) and radiation efficiency of 96% centered at 26 GHz as resonating frequency. The cross-slot which is done on the ground plane enhances the coupling to the Dielectric Resonator Antenna and achieves maximum power radiation along the broadside direction. The slot dimensions are further optimized to achieve the desired impedance match and is also compared with that of a single rectangular slot. The designed antenna can be used for the higher frequency bands of 5G from 24.25 GHz to 27.5 GHz. The mode excited here is characteristics mode of TE_1Y1_. The antenna designed here can be used for indoor small cell applications at millimeter wave frequency band of 5G. High gain and high efficiency make the DRA designed here more suitable for 5G indoor small cells. The results of return loss, input impedance match, gain, radiation pattern, and efficiency are shown in this paper.

## 1. Introduction

To address the diversified requirements from the envisioned 5G usage scenarios, 5G needs access to “high”, “medium”, and “low” level of frequencies. The sub 6 GHz and millimeter wave 30 GHz band (e.g., 24.25 GHz–29.5 GHz and 37 GHz–43.5 GHz) will be most populated frequency bands for 5G. The base station antennas to be used for millimeter wave frequency bands must support high data rate transmission and high efficiency. 5G millimeter wave transmission upgrades to low latency transmissions with high data rate. Microstrip patch, dielectric resonator, and many such antennas have been used in millimeter wave frequency bands, but dielectric resonator antenna (DRA) has gained more popularity because of light weight, small size, zero surface wave loss, and metallic losses. It can achieve wider impedance bandwidth and gain compared to metallic antennas like as micro strip patch antenna. Metallic antennas like microstrip patch antenna has maximum metallic losses but ceramic based dielectric resonator antennas has minimum metallic losses at millimeter wave frequencies [1]. Dielectric resonator antennas (DRAs) are the most suitable candidates to replace the conventional radiating elements at millimeter wave frequencies and especially for indoor applications of millimeter wave frequency bands [2]. DRAs do not have conduction losses and are importantly characterized by high radiation efficiency when get excited with desired radiating mode [3,4]. In aperture coupled technique the slot dimensions made on the ground plane can have the effects of impedance and capacitance as the DRA is placed over the metallic ground plane [5].The physical dimensions of a Dielectric Resonator is the function of its dielectric permittivity and loss tangent of the material used. So, the actual dimension of a DRA can be controlled to its minimal with larger Permittivity value range from 10 to 100. The resonant mode used depends on the geometry of the resonator and the required radiation pattern [6]. The Rectangular Dielectric Resonators have practical advantages over other shapes. Further, for a given resonant frequency, two aspect ratios of a rectangular DRA (height/length and width/length) can be chosen independently. Maximum impedance bandwidth can be achieved using a impedance match between the connector and the DRA [7]. The major advantage of using the rectangular DRA is characterized by three independent geometrical dimensions along three different coordinate axis, and the height of the DRA is 𝑑, which enhances maximum flexibility in the physical dimensions of rectangular DRA while compared to the cylindrical DRA [8,9]. A Dielectric Resonator can be excited through a strip line, Aperture, Coaxial, or substrate integrated waveguide techniques [10,11]. Using a proper excitation or feed technique, a dielectric resonator structure can act as a radiator at desired resonating frequencies. It has to notice that, for any given or desired resonant frequency, the actual size of a dielectric resonator is inversely proportional to its relative permittivity of the constitutive material. An average permittivity of 10 has better impedance bandwidth in DRAs [12]. The essential principle of operation of dielectric resonator is comparable to that of the cavity resonators. Based on such conventional design approaches the most popular radiating dielectric resonators are the cylindrical and the rectangular ones [3,13]. The aspect ratio and Q factor of DRA can be compared with Figure 1. An indoor small cell base station requires highly efficient antennas with lowest impedance mismatch. The complexity of the antenna design at millimeter wave frequency band is high because of impedance mismatch and small size. Dielectric resonator antennas offer wide bandwidth and high efficiency at millimeter wave frequency and which enhances the signal strength to overcome reflection losses and throughput in channel [14]. The use of circular patch as feed has been most popular as shown in previous work [15,16] but the measured gain at low frequencies is quite low. Moreover, a circular patch has been most convenient way to feed the antenna. As metallic antennas has losses associated with its metallic properties. An analytical study comparing a microstrip patch and dielectric resonator antenna has been shown by Guha and Kumar [8]. Using all basic feed mechanism. A dielectric resonator can exhibit as a magnetic dipole under basic feeding mechanism. The impedance variation in different feeding techniques helps to excite the DRA with better reflection coefficient and with desired mode of excitation. Circular patch antennas have been used as more convenient antennas because of low profile characteristics. In this paper an aperture coupled technique has been used fed by a microstrip circular patch antenna placed on the other side of the substrate. The impedance bandwidth of a Dielectric Resonator is the function of materials permittivity and Length to Height ratio. Because of Advantages like low loss, small size, wide bandwidth, and easy of excitation the dielectric resonators are used at mm wave transmissions. The most conventional feed technique in DRAs is using a slot over the ground plane. The impedance match allows the antenna to deliver maximum radiating power along the desired direction and the slot apertures made on the ground plane excites the required resonant modes of the DRA [17,18]. The antenna designed here has used a cross type slot over the ground plane to enhance the gain of the DRA. This gain enhancement can be considered as a better feed mechanism compared to other feeding schemes. Small cells in 5G needs high gain and wideband antenna system for indoor cellular services. The throughput and efficiency of the channel can be enhanced using the high gain and efficient antenna designs. The interference and internal reflections reduce the signal strength, so for a high gain antenna system can be a better system. Indoor small cells need high gain antenna performances as the reflections across the walls will generate maximum attenuation.

The design proposed in this paper which can be used for indoor 5G applications in the frequency band of 24.25 GHz–27.5 GHz. This novel design method can also distinguish between the Dielectric Resonator Antennas to other conventional antennas as microstrip patch. The simulation design and study are carried out using High frequency structure simulator (HFSS). The optimization results of slot dimensions, input impedance to the antenna and DRA dimensions are also presented here. Here the simulation work is carried out using HFSS and DRA dimensions are calculated using Mat Lab. In Section 2 the antenna design and basic calculations are expressed with study of feed dimensions, aperture coupled mechanism and gain enhancement. Section 3 explains about the optimization study and analysis of DRA dimensions and radiated field effects and its comparison of rectangular slot with the cross slot. The limitations of such antenna design are also discussed in the Section 4 of this article. A manual fabrication and glue technique used in fixing the DRA over the substrate need high accuracy and proficiency.

## 2. Antenna Design and Analysis

A ceramic ECCOS-TOCK Hik material type Dielectric Resonator antenna is used with permittivity *ε_r_* = 10 and loss tangent 0.002 over a substrate of Roger RT/Duroid 5880 with permittivity 2.2 and loss tangent 0.009. The ground plane is above the substrate and a micro strip patch is used as feed and is placed below the substrate. The slot is created via the ground plane over which the DRA is placed. The slot dimensions are calculated with respect to the resonating wavelength 11.52 mm and are further optimized to match with required input impedance of 50 Ohm. The calculated dimensions of the DRA are a = 2.9 mm, b = 2.6 mm and d = 1.4 mm which is shown in Figure 1. The calculated dimensions of the substrate are Sub_L_ = 5.76 mm, Sub_W_ = 5.76 mm, Sub_h_ = 0.254 mm. The feed line dimensions calculated are L_1_ = 0.63 mm, L_1W_ = 0.15 mm. The position of the DRA can be moved either along *x* or *y* direction to achieve an efficient coupling. Here a high-quality factor (Q) vale of 14 has been achieved theoretically considering the permittivity of DRA as 10. The resonant frequency of the DRA is proportional to є_r_^−0.5^, so for a wide range of permittivity values can be used to resonate the antenna at required frequency bands. Figure 1 represents the DRA design method. The theoretical calculations for resonating frequency of a rectangular dielectric resonator antenna are shown in Equations (1) and (2).
(1)$$k_{x}\times \tan\left(k_{x}d/2\right)\;=\;\sqrt{(\varepsilon_{r}-1})k_{0}^{2}-k_{x}^{2},$$
where
$$k_{0}\;=\;\frac{2\pi }{\lambda_{0}\;}\;=\;\frac{2\pi f_{0}}{c}$$
$$k_{y}\;=\;\frac{m\pi }{b}$$
and
$$k_{z}\;=\;\frac{n\pi }{d},$$
and
(2)$$k_{x}^{2}+k_{y}^{2}+k_{z}^{2}\;=\;\varepsilon_{r}\times k_{0}^{2},$$
where c is velocity of light, *ε_r_* is relative permittivity of DRA, *k*_0_ is free space wave number, m and *n* are called as half-wave field variations along the y and z directions, respectively. The symbols *k_x_*, *k_y_*, and *k_z_* represent the wave numbers in the *x*, *y*, and *z*-directions, respectively, and a, b, and d indicates the dimensions of DRA which are proportional to the square root of dielectric constant values of the ceramic based DRA.

The measured dimensions of the DRA are calculated using Mat lab and the simulated and optimized details of DRA dimensions are shown in Table 1. Figure 2 shows the top view and bottom view of the antenna design. The resonant frequency of the rectangular DRA can be found from Equation (1). The rectangular dielectric resonator offers three degree of freedom as it has three coordinate axes with respect to length, width, and height of the DRA. *k_X_*, *k_Y_*, and *k_Z_* are the three coordinate axis wave numbers along *x*, *y*, and *z* direction of the DRA. A rectangular DRA can have three different characteristic’s modes which are called as the fundamental modes of DRA TE_X11_, TE_1Y1_, and TE_11Z_ [19].

Ansys HFSS is used here for design and simulation of the antenna. All the boundary conditions are satisfied to create a perfect electric field environment. The unit cell design is placed at the center of the coordinating axes which is *x* = 0 and *y* = 0. The DRA is placed above the substrate so the height along the z axis is the substrate height. The perfect electric field ground plane is generated to match with the potential differences of the antenna. The DRA dimensions that are optimized further to match with the required impedance bandwidth and the calculation for the quality factor of the DRA are shown in the Table 2. Here both the length and the width of the DRA are kept similar and the height of the DRA is optimized to excite the DRA under the characteristic’s mode. The aspect ratio of the DRA can further be optimized with the varying dimensions height to length ratio. The mode of excitation either can be TE (Transverse Electric) or TM (Transverse Magnetic) depending upon the physical dimension and electric field distribution of the DRA. The coupling of radiated field between the patch and the DRA depends on the physical dimensions of the cross slot made on the ground plane. Table 2 has the details of aspect ratio, quality factor (Q) and impedance bandwidth of the antenna. The radiation Q-factor is then found by determining the radiated power and stored energy. Further the quality factor can be analyzed with the dimension ratio of the DRA and the aspect ratio requirements for wider bandwidth. The substrate and the ground plane dimensions are (5.76 mm × 5.76 mm). The slot is made at the center of the ground plane which is placed over the substrate.

### 2.1. The Metallic Circular Patch as the Feed to DRA

The circular patch element is used to feed the DRA across the cross slot made over the ground plane. The coupling of electric fields across the slot depends on the power radiated by the patch. Figure 3a shows the fields radiated across the patch and Figure 3b shows the fields radiated over the DRA. The radius of the patch is further optimized to check with the input impedance of the DRA. Both the DRA and the patch is centered at the same point and the electric field radiated by the patch element is easily coupled to the DRA.

The power radiated by the circular patch depends upon the slot impedance as well as the load impedance at the connector. The difference between the radiated power and the input power will measure the radiation efficiency of the DRA. Here, the slot impedance and the load impedance (impedance offer at the connector to the microstrip line is maintained to minimum such that maximum power can be delivered to the radiated circular patch. Figure 4 shows the magnetic dipole moment distribution over the patch. The power radiated by the circular patch can be used to measure the conductance and the directivity of the antenna. As the circular patch is placed at the center of the substrate and at position *z* = 0, maximum power will be radiated to the DRA through the slots made on the ground plane.

### 2.2. Aperture Coupling Mechanism and Calculations

The DRA is placed over the ground plane where slot is been made. The slot dimensions as slot length and slot width can be calculated as the desired resonant frequency of the antenna. The DRA is placed exactly over the slot apertures above to a height of 0.254 mm which is the thickness of the substrate. The metallic circular patch is placed on the other side of the substrate. The dimensions of both the ground plane is maintained same as the substrate dimensions and calculated in terms of wavelength. The optimization study of the partial ground plane effect in done in the next part of the paper. Equations (3) and (4) represents the equations for calculating the slot dimensions.

The slot dimensions can be calculated as

Slot Length,
(3)$$S_{L}\;=\;\frac{0.4\lambda_{0}}{\sqrt{\varepsilon_{eff}}}$$
where $\varepsilon_{eff}$ is called as effective permittivity which is calculated as
$$\varepsilon_{eff}\;=\;\frac{\varepsilon_{r}+\varepsilon_{s}}{2}$$
where $\varepsilon_{r}$ and $\varepsilon_{s}$ are the dielectric constants of the DRA and the substrate, respectively.

Slot Width,
(4)$$S_{w}\;=\;0.2\times S_{L}$$

The calculated vales slot length and slot width are at the resonating frequency of 26 GHz as shown in Table 3. The electric fields radiated into the DRA depends on the calculated aperture dimensions and the amount of coupling depends on electric field distributed over the DRA. Equation (5) represents the coupling factor from feed line to DRA through a slot on the defined ground plane.

The coupling factor (C) of the DRA can be expressed as
(5)$$C\;=\;\int_{v}\left[\left(E_{DRA}\;\cdot Js\right)dV\right],$$

Here *E_DRA_* is the electric field vector distribution over the transmission line and *Js* is a uniform current source. The distributed surface current need to be controlled over the thickness of the dielectric substrate. The coupling factor is proportional to the slot dimensions on the ground plane, as the electric field distribution over the DRA generates an electric and magnetic dipole moment. There are slight variations in the simulated and theoretical values of slot dimensions made on the ground plane. The calculated values are used to optimize the antenna design parameters to achieve the maximum impedance bandwidth. The slot dimension calculations are shown in Table 3. The slot length and width dimensions for the cross slot is uniform and is placed at the center of the coordinate axis. The slot impedance is here uniform to that of characteristics impedance of the feed which helps in enhancing the radiation efficiency of the DRA. The electric field distribution over both XZ and XY plane is shown in Figure 5. The excited characteristics mode here is TE_1Y1_, the cross slot excitation has increased the gain of the DRA without changing its mode of excitation. The fundamental characteristics mode of excitation here matches with the 50 Ohm input impedance of the DRA. The DRA here is linearly polarized and can be used for unidirectional indoor 5G applications because of its high gain and efficiency. The impedance bandwidth improvement from rectangular slot to cross slot is about 0.8 percentage which is a bandwidth of 0.22 GHz. Figure 6 shows the electric volume magnitude in dB scale.

The gain enhancement occurs by changing the rectangular slot to a cross slot, where a cross slot fed DRA acts as a magnetic dipole. There a slight improvement in the gain values has been achieved from rectangular to cross slot over the ground plane. The cross slot excites the DRA with higher electrical energy coupling resulting in enhancing the gain of the DRA. The impedance variation occurs when a rectangular slot is replaced with a cross slot over the ground plane. The slot dimensions are responsible for the coupling between the DRA and the circular microstrip patch. The gain improvement of 0.35 dB has been recorded here. The cross slot enhances gain with minimum cross polarization. The rectangular slot is placed at the upper edge of the DRA, but the cross slot is placed at the center of the DRA. These slot positions are optimized locations and are studied and fixed to resonate at the desired resonating frequency of the antenna. The study of mode excitation and fields is shown in Figure 5 and Figure 6.

## 3. Evolutionary Study and Analysis

The design of the final antenna has been done from a conventional way of feeding a dielectric resonator antenna which is shown in Figure 2. The DRA is placed over a rectangular slot made on the ground plane. The ground plane is placed above to the substrate and a micro strip line is placed at the other side of the substrate. The gap of 0.254 mm which is the thickness of the substrate is maintained between the micro strip feed line and the rectangular slot or DRA on the ground plane. The length and width of the micro strip line is optimized to match with the characteristic’s impedance of the antenna. A general input impedance of 50 ohm is observed to excite the DRA under fundamental modes. The magnitude of electric field distribution over the two different slots are also shown in Figure 7c,d. The cross slot has been prepared with two narrow rectangular slots of equal length and width. The optimization study of different slot dimensions with impedance bandwidth has also presented here. The optimization study is carried out for the rectangular slot and the cross slot over the ground plane. The characteristics mode analysis also has been done at different slot length and width dimensions. For characteristics mode of TE_1y1_ the electric field across the slots follows the similar radiation pattern as per both E-plane and H-plane, which minimizes the back-lobe radiation in the DRA. Figure 7 shows the evolution process of final DRA design from using a rectangular slot to a cross slot and Figure 8 shows the electric field density in dB scale. The uniform slot dimensions also reduce the cross polarized power in the antenna. The feed of either a cross slot or a rectangular slot make the DRA act like a magnetic dipole. As the DRA is placed directly above the ground plane with a cross slot helps also to reduce the back propagation and cross pol power of the antenna. The cross slot offers a minimum cross pol in the design.

In millimeter waves the dimensions of substrate and ground plane are calculated in terms of wavelength of desired resonating frequency and controlled to uniform the field distribution and fringing fields over the DRA and the ground plane. The Electric fields over the metallic patch are radiating outwards and is maximum at the feed point as a conventional radiator. The DRA works like a magnetic dipole under is characteristics mode of excitation as shown in Figure 7c,d. The characteristics of both the rectangular and cross slot are studied and its performance analysis is recorded in Table 4. The cross slot has achieved high gain and wide bandwidth and high isolation as compared to the rectangular slot on the ground plane. Figure 9 shows the reflection coefficient of both the rectangular and cross slot and Figure 10 shows the corresponding input impedances at the resonating frequencies. The dimensions of the slot apertures for both rectangular and cross slot are remained uniform. The center of the cross slot and rectangular slot are coincided with the DRA and the coordinate axes. The co and cross pol radiation pattern for both the rectangular and cross slot are shown in Figure 11 and Figure 12.

The simulated gain values for both rectangular and cross slot in both co and cross polarization is shown in Figure 11 and Figure 12 in both YZ plane and XZ plane, respectively. There is non-uniform radiation pattern with low cross pol power in both the planes. The maximum radiation is along the broadside direction of the antenna. The simulation study has been carried including all the antenna parameters as reflection coefficient, gain, and radiation pattern for both the rectangular and cross slot on the ground plane.

The performance of DRA fed by both rectangular slot and cross slot is shown in Table 4. The gain and bandwidth both have been improved by using a cross slot than a rectangular slot. This gain enhancement is observed without changing the feed dimensions. There is an improvement of 0.22 GHz in bandwidth which is 0.8% from rectangular slot to cross slot. There is also an improvement in gain values. This optimization study has helped in finding the maximum gain, efficiency, and bandwidth of the antenna.

## 4. Results

The simulated and measured reflection coefficient for cross slot is shown in Figure 13 and the corresponding input impedance at different patch radius is shown in Figure 14. The characteristics impedance of 50 Ohm has been maintained near the patch radius, which has helped in achieving.
Delivering maximum power to the antenna which enhances the radiation efficiency further.Minimum cross pol power in both the E and H plane of the antenna.

The patch radius is varied with different radius to achieve the desired impedance bandwidth. The simulation study of the patch radius is also carried out and its reflection coefficient is shown in Figure 15. At patch radius 2.2 mm it has achieved maximum isolation of −32.4 dB and other values isolation data is shown in Table 4 at different patch radius. The input impedance at 26 GHz with patch radius 2.2 mm is 49 Ohm which matches with the characteristic’s impedance of the antenna. The simulated and measured reflection coefficient isolation is −44.82 dB and −29.23 dB, respectively. The DRA dimensions are calculated and are optimized in terms of the aspect ratio with respect to the Q factor. A bandwidth of 1.12 GHz is suitable to use this antenna for indoor 5G small cells. Here the DRA is excited under the characteristic’s mode TE _1Y1_. The aspect can be optimized by changing the height of the DRA. Figure 16 shows the reflection coefficient of DRA at different DRA heights.

The physical dimension of the DRA is very small, which makes several fabrication errors and destabilizes the radiation characteristics of the antenna. Here, the input impedance of the strip line to the patch is varied based on the impedance offed by the connector. Moreover, the characteristics impedance is well matched which is around 49 Ohm and has offered maximum radiation efficiency. Figure 17 and Figure 18 shows the optimization of feed length and width of the microstrip line and its corresponding input impedance variation. Table 5 represents the reflection coefficient at different dimensions of the cross slot. The cross slot has been made from a rectangular slot. The variation in the slot structure shows that the cross pol of cross slot is higher than that of rectangular slot maintain a non-variable input impedance. The uniform variation in the slot dimension varies the impedance values.

Different slot dimensions changes the voltage across the terminal load and is mode dependent on the impedance appears at the feed terminal. The Table 6 shows the feed length and the impedance bandwidth isolation of the DRA. The Maximum isolation of −52.07 dB is observed at the slot dimensions of 1.2 mm length and 0.8 mm width. The dimensions of cross slot are similar and is placed at z = 0. At first the rectangular slot is studied and to improvise the characteristics parameters a cross slot has been made on the ground plane. The change in slot structure has not much change the cross-pol power but has enhanced the gain and bandwidth of the antenna. Figure 19 shows the fabrication process and the measurement of antenna parameters in the anechoic chamber.

Figure 20 represents the gain and efficiency measured at the desired resonating frequency. The DRA exhibits a higher efficiency of 96 percentage and gain improvement to 10 dB. Figure 21 and Figure 22 represents the gain of DRA both in E plane and H plane. The simulated isolation in cross pol power in E plane is −45.62 dB and in H plane is −49.73 dB. The measured values of isolation in cross pol −43.64 dB in E plane and −49.12 dB in H plane. The simulated and measured gain of the DRA is 10.57 dB and 8.04 dB, respectively. The cross slot in the ground plane has reduced the cross-pol power and has enhanced the gain of the DRA. The cross pol reduces the back-lobe radiation of the antenna. There is slight shift in the cross-pol minima by 10 degree in the measured results. Similarly, the co pol and cross pol power in H plane is shifted by 10 degree. The recorded efficiency of the DRA is 96 percent, as the slot impedance is 49 ohm which is very close to the characteristic’s impedance of 50 ohm. So maximum power has been delivered to the antenna enhancing its gain and efficiency. In Table 7, the previous work on DRA has been recorded and compared with this work at millimeter wave frequency bands. This singly fed DRA design has more advantages compared to the previous work in terms and gain and efficiency. The DRA designed here can be used as an efficient radiator for 5G indoor small cells. The antenna is linearly polarized and has maximum radiation along the broadside direction which can help in minimizing the path loss component of the propagation. The performance parameters as bandwidth, gain, efficiency, and radiation pattern are shown here.

In Table 7 the gain, bandwidth, electrical dimensions, and efficiency of the designed antenna is compared with other antennas referred. With permittivity of 10 the DRA has achieved a higher gain value compared to other antennas. The quarter wavelength dimensions of the antenna are electrically larger at desired resonating frequency. Generally, the circular metallic patch used here offers an upward electric field coupling to the DRA through the cross slot with low surface impedances. The reactance value of the antenna is very low and is matched with the input impedance resulting in a broadside radiation.

## 5. Discussion

Using a simple novel method of aperture coupling a Dielectric resonator antenna with permittivity of 10 has achieved a gain of 8.04 dB. A non-resonating circular patch antenna has been used as a feed to the ceramic based dielectric resonator antenna. Cross polarization power is also minimized with the slot dimensions made on the ground plane. The difficulty was high in the fabrication of the antenna and its placement on the ground plane. The design will be further developed to large array patterns to achieve higher gain and efficiency. There is a shift in the resonating frequency from simulation study to measurement study because of

Difficulties in the placement of the DRA on the slot aperturesUse of conductive glue to fix the DRA on the ground plane.

Whereas the impact on radiation efficiency was less as the measured cross polarization power was less.

3.Fabrication error.

## 6. Conclusions

Dielectric Resonator Antennas can be preferred because of wide bandwidth and high efficiency at millimeter wave frequency bands. 5G band 30 GHz from 24.3 to 27.8 GHz band is widely used and can be used for indoor cellular applications. All the design parameters were calculated using MATLAB and all boundary conditions for simulation environment were achieved here. The measurement conditions were satisfied between the DRA and the receiver antenna. The DRA is linearly polarized and is excited under the characteristic’s mode TE_1Y1_. An enhancement in gain and bandwidth has been also done by using a cross slot aperture in ground plane. A singly fed DRA proposed here has achieved high gain (10.57 dB simulated and 8.04 dB Measured), high efficiency (96% Measured and 98% Simulated), and wide bandwidth (1.12 GHz) makes it more suitable antenna for indoor millimeter wave 5G small cell applications.

## Figures and Tables

**Figure 1 sensors-21-02694-f001:**
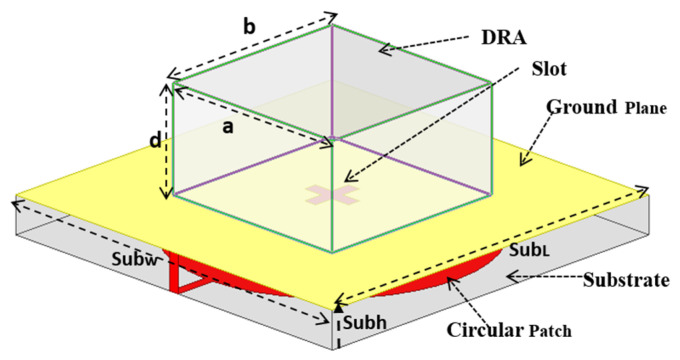
Dielectric Resonator Antenna fed by a Circular Patch.

**Figure 2 sensors-21-02694-f002:**
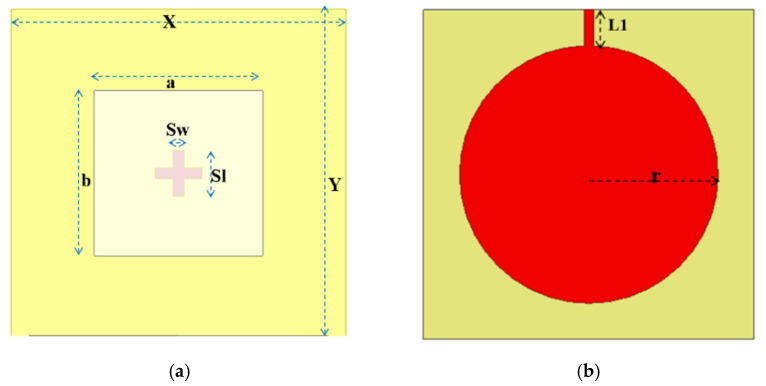
Aperture Coupled Dielectric Resonator Antenna: (**a**) Top View showing the Cross Slot; (**b**) Bottom View showing the circular patch.

**Figure 3 sensors-21-02694-f003:**
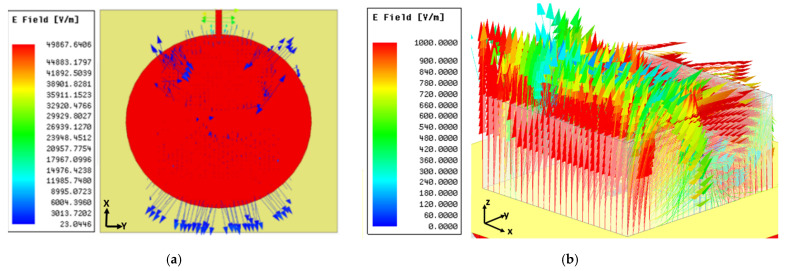
Electric field distribution over (**a**) metallic circular patch (**b**) DRA.

**Figure 4 sensors-21-02694-f004:**
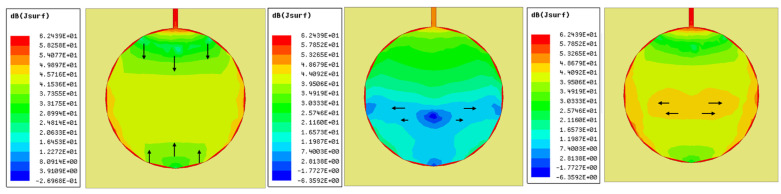
Electric field distribution over DRA radiating like a dipole moment.

**Figure 5 sensors-21-02694-f005:**
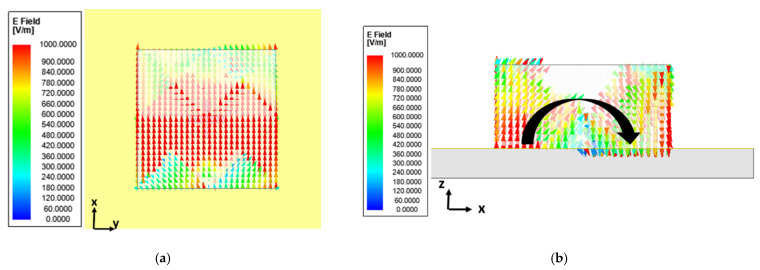
Electric field distribution over DRA (**a**) XY Plane (**b**) XZ Plane.

**Figure 6 sensors-21-02694-f006:**
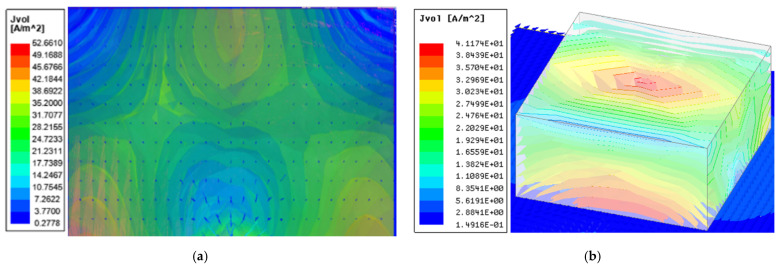
Electric field magnitude distribution over DRA (**a**) Field density front view (**b**) 3D View.

**Figure 7 sensors-21-02694-f007:**
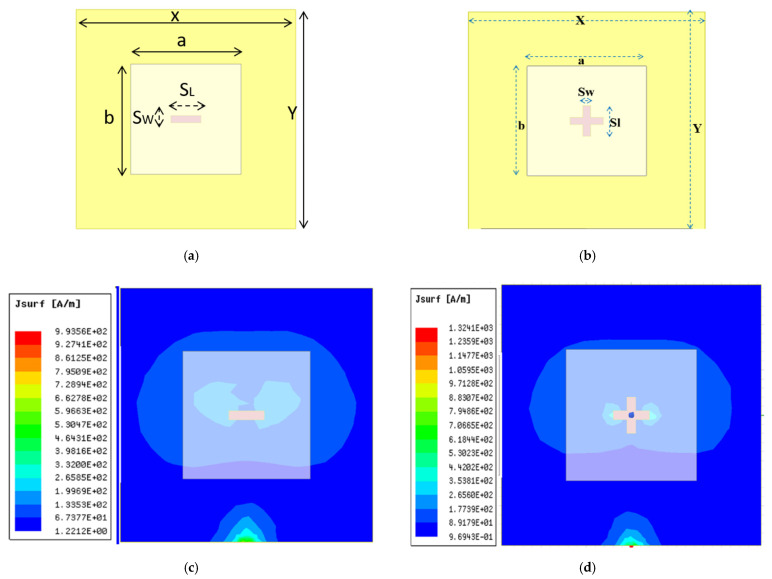
Evolution of the antenna design: (**a**) Micro strip line fed with rectangular slot DRA; (**b**) Circular patch fed cross slot DRA (**c**) Surface current density over a rectangular slot (**d**) Surface Current density over a cross slot.

**Figure 8 sensors-21-02694-f008:**
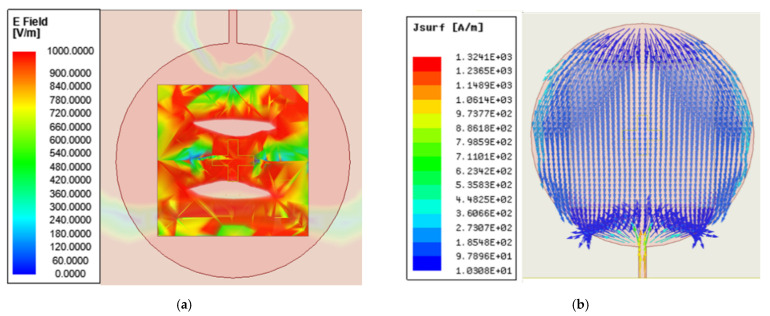
(**a**) Electric field distribution over the circular patch in XY Plane (**b**) Electric field distribution over the circular patch in XY plane.

**Figure 9 sensors-21-02694-f009:**
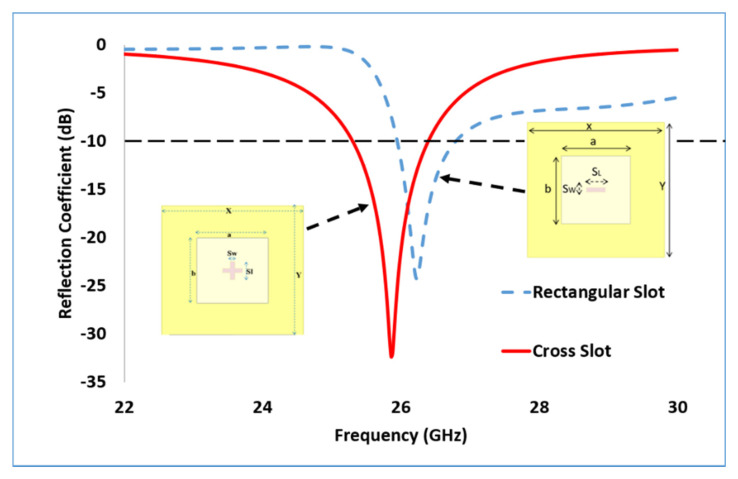
Simulated Reflection coefficient (dB) for different slot structures.

**Figure 10 sensors-21-02694-f010:**
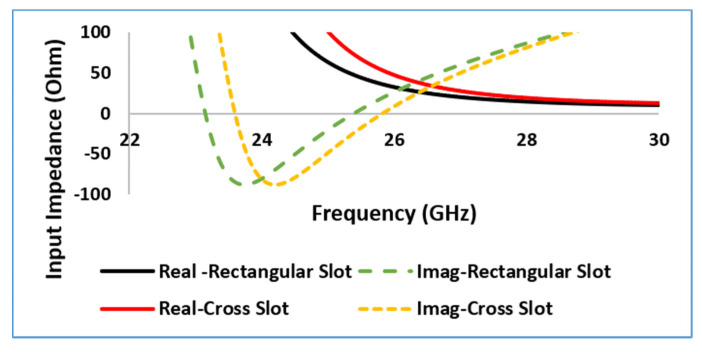
Input Impedance (ohm) at different slot structures.

**Figure 11 sensors-21-02694-f011:**
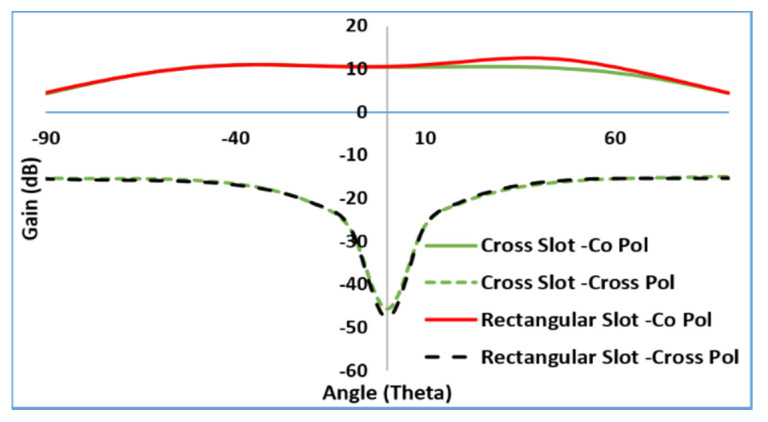
Simulated Radiation pattern in YZ Plane for different slot structures.

**Figure 12 sensors-21-02694-f012:**
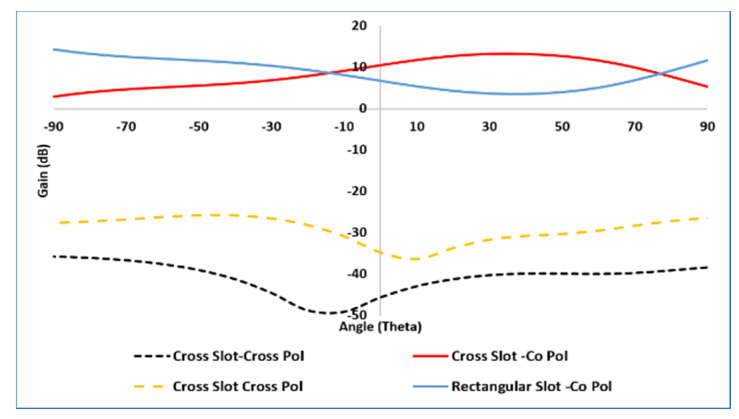
Simulated Radiation pattern in XZ Plane for different slot structures.

**Figure 13 sensors-21-02694-f013:**
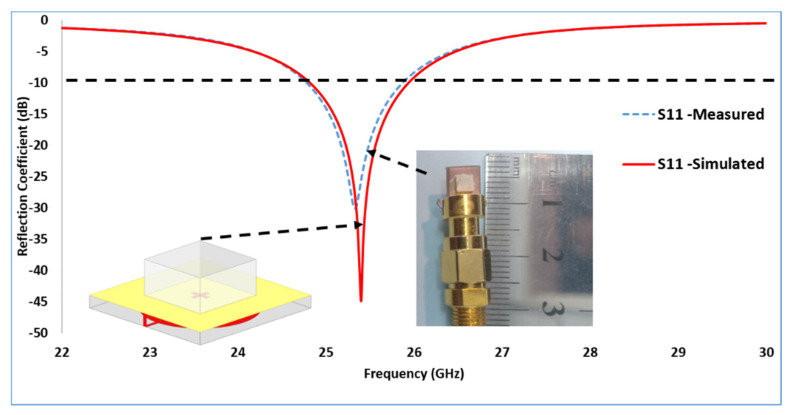
Reflection coefficient (dB) vs. frequency (GHz).

**Figure 14 sensors-21-02694-f014:**
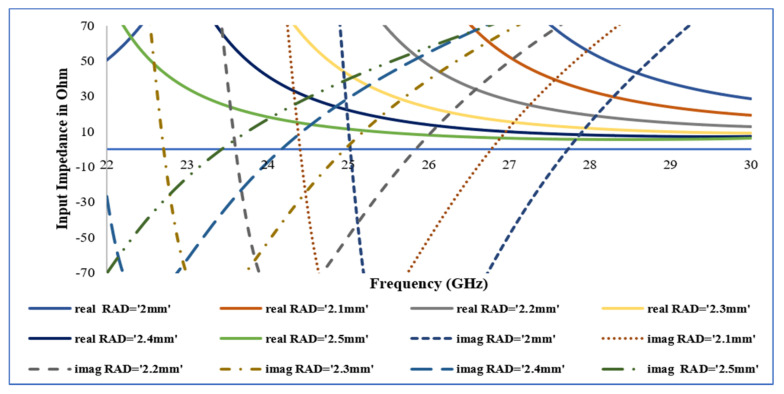
Variation of Characteristic’s input impedance in ohm at different radius of circular patch in mm.

**Figure 15 sensors-21-02694-f015:**
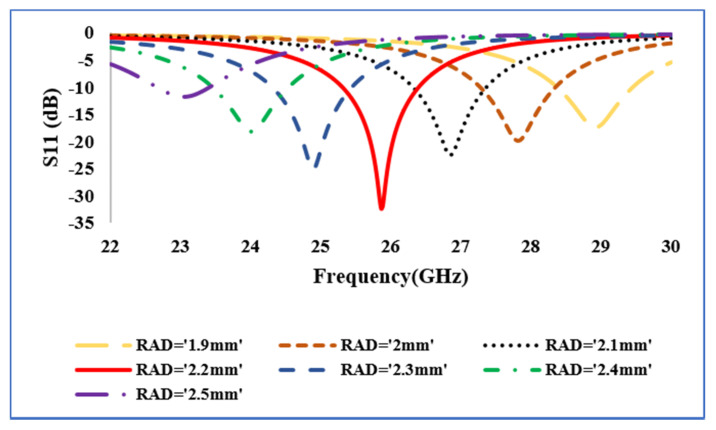
Reflection coefficient in dB at different radius values of the circular patch in mm.

**Figure 16 sensors-21-02694-f016:**
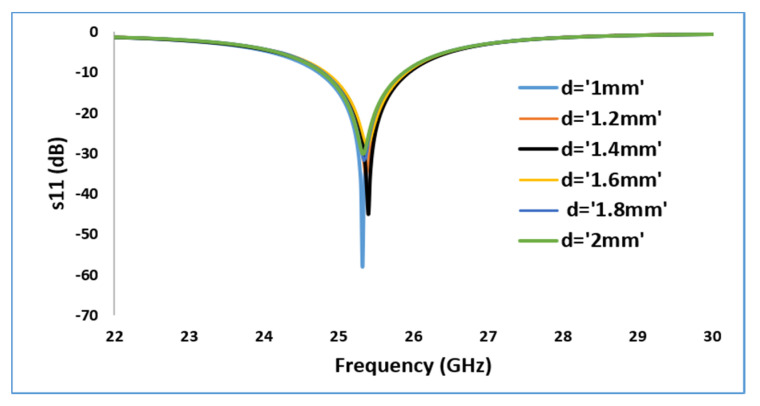
Reflection coefficient in dB at different radius values of the circular patch in mm.

**Figure 17 sensors-21-02694-f017:**
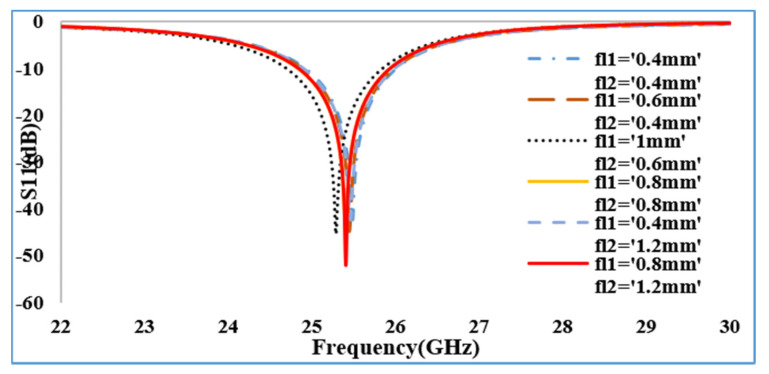
Reflection coefficient (dB) vs. Frequency (GHz) at different feed length (fl).

**Figure 18 sensors-21-02694-f018:**
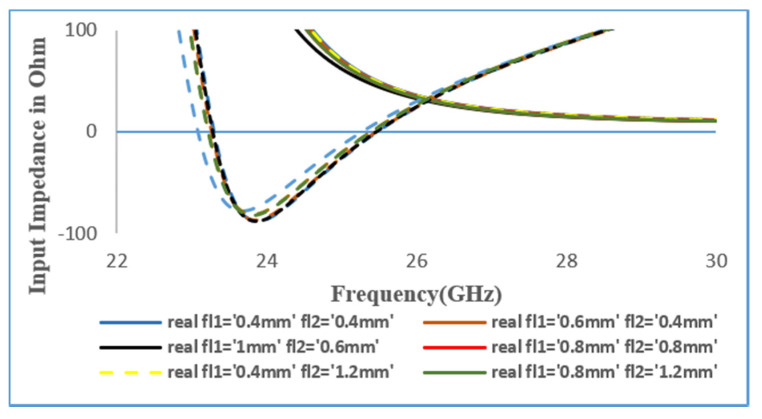
Input Impedance of DRA at different feed dimensions.

**Figure 19 sensors-21-02694-f019:**
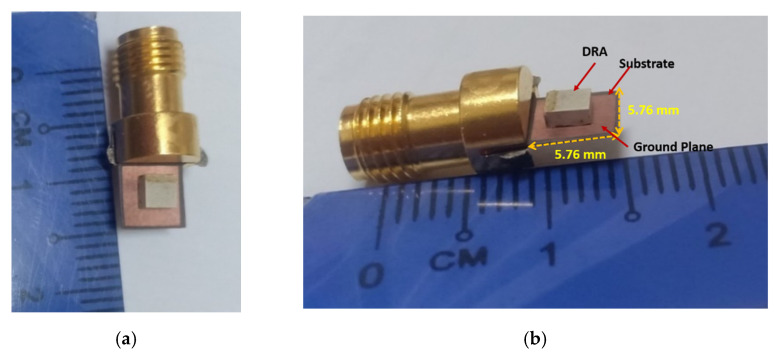
Fabricated Rectangular DRA at resonating frequency 26 GHz (**a**) Rectangular DRA Top View; (**b**) side view (**c**) Measurement in anechoic chamber (**d**) Cross slot aperture.

**Figure 20 sensors-21-02694-f020:**
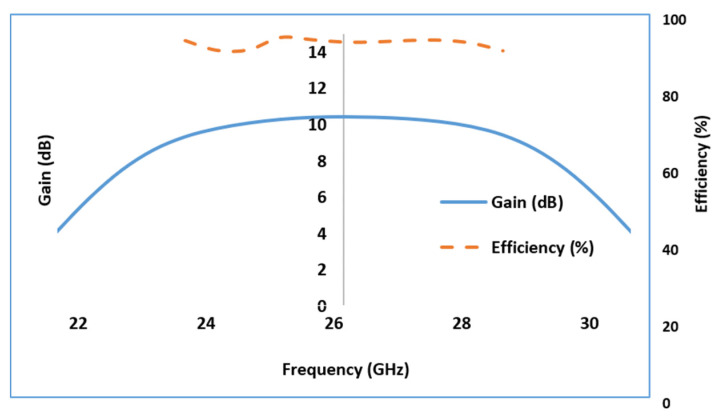
The measured gain and efficiency of the DRA.

**Figure 21 sensors-21-02694-f021:**
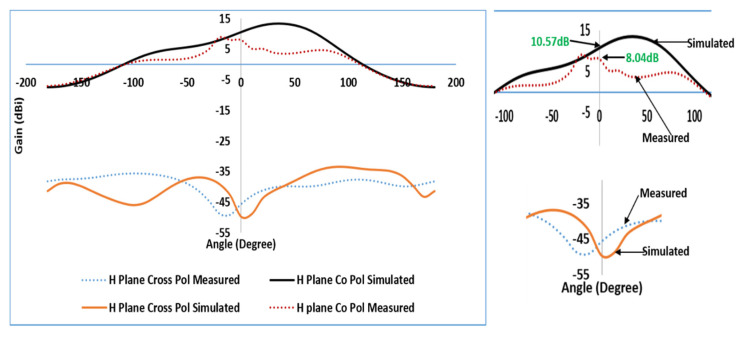
Radiation pattern XZ plane.

**Figure 22 sensors-21-02694-f022:**
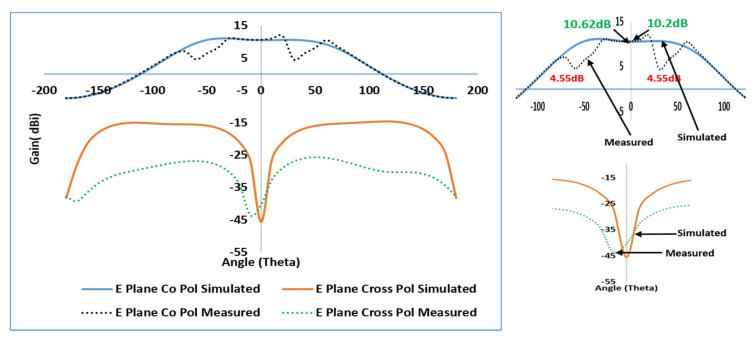
Radiation Pattern YZ plane.

**Table 1 sensors-21-02694-t001:** Antenna Design Specifications. (DRA: Dielectric Resonator Antenna).

Antenna Parameters	Parameter Details	Values in mm
a	DRA Width	2.9
b	DRA Length	2.9
d	DRA Height	1.4
S_w_	Slot Width	0.2
S_L_	Slot Length	0.8
X	Ground Plane Width	5.76
Y	Ground Plane Length	5.76
Subh	Substrate Height	0.254
SubL	Substrate Length	5.76
SubW	Substrate Width	5.76
r	Radius of Patch	2.25
L1	Patch Feed Length	0.63
L1W	Patch feed Width	0.15

**Table 2 sensors-21-02694-t002:** Antenna Design Calculations for aspect ratio and bandwidth.

є_r_	a (mm)	a/b (mm)	d/b (mm)	b (mm)	d (mm)	Q	Bandwidth (%)
10	2.9	1	0.48	2.9	1.4	14.13	5 (Simulated) 4 (Measured)

**Table 3 sensors-21-02694-t003:** Slot dimension Calculations.

Resonating Frequency f_r_ (GHz)	DRA Permittivity $\varepsilon_{r}$	Substrate Permittivity $\varepsilon_{s}$	Effective Permittivity $\varepsilon_{eff}$	Slot Length $S_{L}$ (mm)	Slot Width $S_{w}$ (mm)
26	10	2.2	6.1	1.86 (Theoretical) 0.8 (Optimized)	0.37 (Theoretical) 0.2 (Optimized)

**Table 4 sensors-21-02694-t004:** Performance Comparison of different slot apertures.

Slot Type	Bandwidth (GHz)	Bandwidth (%)	S11 Isolation (dB)	Gain Simulated (dB)
Rectangular Slot	0.84 GHz (25.96–26.8 GHz)	3.23%	−24.18 dB	6.8 dB
Cross Slot	1.06 GHz (25.32–26.38 GHz)	4.07%	−32.35 dB	8.04 dB

**Table 5 sensors-21-02694-t005:** Return Loss (dB) at different slot dimensions.

Slot Length (fl1,fl2)	Resonating Frequency (GHz)	Return Loss (S11) (dB)
0.4,0.4	28.98	−30.86
0.6,0.4	27.82	−37.50
1.0,0.6	26.84	−45.36
0.8,0.8	25.86	−38.04
0.4,1.2	24.92	−39.78
0.8,1.2	24.06	−52.07
0.4,0.4	23.02	−30.86
0.6,0.4	28.98	−37.50

**Table 6 sensors-21-02694-t006:** Return Loss (dB) for different patch radius.

Patch Radius in mm (RAD)	Resonating Frequency (GHz)	Return Loss (S11) (dB)
1.9	28.98	−17.09
2.0	27.82	−19.88
2.1	26.84	−22.51
2.2	25.86	−32.35
2.3	24.92	−24.97
2.4	24.06	−18.14
2.5	23.02	−11.07
1.9	28.98	−17.09

**Table 7 sensors-21-02694-t007:** Performance Analysis of the proposed work.

Ref No	Frequency GHz	DRA Shape	Permittivity *ε_r_*	Bandwidth %	Electrical Dimensions	Gain dB	Efficiency %
[20]	35 GHz	Cylindrical	10	15.6	0.14λ_0_ × 0.12 λ_0_	6.9	95
[21]	36.22 GHz	Rectangular	10.2	8.6	0.24λ_0_ × 0.3λ_0_ × 0.24λ_0_	5.51	95
[22]	24 GHz	Rectangular	10	3.74	0.38λ_0_ × 0.51λ_0_ × 0.24λ_0_	5.9	Not Mentioned
[23]	60 GHz	Rectangular	12.6	6.1	0.2λ_0_ × 0.2λ_0_ × 0.2λ_0_	6.0	98
[24]	35 GHz	Cylindrical	10.2	11.0	0.4λ_0_ × 0.4λ_0_ × 0.07λ_0_	5.5	88
[25]	26 GHz	1*2 MIMO Rectangular DRA	10.2	7.3	0.39λ_0_ × 0.39λ_0_ × 0.11λ_0_	7.1	Not Mentioned
[26]	32 GHz	Grid DRA	17	5.31	0.3λ_0_ × 0.3λ_0_ × 0.1λ_0_	6.4	89
Proposed	26 GHz	Rectangular	10	4.07	0.25λ_0_ × 0.25λ_0_ × 0.12λ_0_	8.04	96

## Data Availability

Not Applicable.

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
