# Peer review of "Circular Patch Fed Rectangular Dielectric Resonator Antenna with High Gain and High Efficiency for Millimeter Wave 5G Small Cell Applications"

_sensors, 2021, doi:10.3390/s21082694_

Round 1

Reviewer 1 Report

Paper claims for the modeling and analysis of rectangular dielectric resonator antenna. Here, the design of dielectric resonator antenna and its performance is shown via simulation and measurement, but further modifications should be needed in terms of performance and contextual flow. In other words, it seems that the manuscript is not yet ready for publication. Detailed comments are as the followings:

  1. Overall, there all numerous flaws in the manuscript. There are many typos and errors in commas and periods, spaces, capitalization, and spelling. Moreover, the numbering of the table is wrong (repeated in Table 4), and the caption of Table 5 is not written. For this reason, the paper is not in perfect condition for publication.
  2. In terms of contextual flow, sentences with the same content are repeated unnecessarily. This can be found especially in the introduction, and the same explanation is repeated over and over again in the results. Please delete redundant descriptions and explain the core content.
  3. In Table 1 and Table 2, the same parameters are shown repeatedly except for the Q factor and bandwidth. This is also redundant and needs to be corrected.
  4. The legend is not clearly visible in all figures. Please modify the picture so that it can be seen clearly.
  5. In the first paragraph of section 1 (line 70), the authors explain that high Q factor can be seen in Figure 1. However, in this part, it seems difficult to check the Q factor simply through Figure 1. Please add a more detailed description in this regard or mark it in Figure 1.
  6. The authors argue that the use of a cross slot improves gain and bandwidth compared to rectangular slot. However, in Table 4, However, there is only a difference of 200MHz in terms of bandwidth, which cannot be seen as a significant improvement.
  7. In Figure 17, the term ‘fl’ is undefined and Figure 20 is not explained in the manuscript.
  8. Please add a comparison of the bandwidth and the electrical size at the center frequency in Table 6. Since the proposed antenna has a large electrical size, it seems that it is necessary to additionally compare the gain in this regard.

Author Response

Dear Editor,

Thank you for allowing a resubmission of our manuscript, with an opportunity to address the reviewers’ comments.

This research article has been submitted to Sensors with manuscript ID Sensors-1162560. We received the decision of the manuscript on 26th March 2021 with reviewers’ recommendations to improve this article. All the review comments have been answered for each reviewer. This article focuses on design of Dielectric Resonator Antenna at millimeter wave frequency bands for 5G applications. A systematic investigation and analysis have been done in designing the antenna. This article will be helpful for the researchers in 5G.      

     We are uploading (a) our point-by-point response to the comments (response to three reviewers in three files), (b) an updated manuscript with “Track Changes” function highlighting indicating changes, and (c) a clean updated manuscript without highlights (PDF main document).

Best regards,

Assoc. Prof. Dr. Mohd Haizal bin Jamaluddin

Wireless Communication Center

School of Electrical Engineering

Universiti Teknologi Malaysia, Johor, Malaysia

Reviewer 2 Report

Abstract appropriately summarize the paper.

 Keywords are appropriate.

1.Introduction

Introduction is concise.

2.Antenna Design and Analysis

Table 2 is unclear. What is the simulation and what is the measurement?

Table 3 is also unclear. 

  1. Optimization Study and Analysis

„The optimization study of different slot dimensions with impedance bandwidth has also presented here. The optimization study is carried out for the rectangular slot and the cross slot over the ground plane. The characteristics mode analysis also has been done at different slot length and width dimensions.“

This chapter would be closer  to evolutionary study and analysis than optimization.

  1. Results and Discussion

The discussion of the results was conducted correctly

Table 4  is unclear.

The last row in Table 4 is unnecessary or incorrect (if f is greater than 2.5 GHz).

  1. Discussion

Two titles have the same number (4. Results and discussion and 4. Discussion). That needs to be corrected. Two chapters on discussion may not be needed.

  1. Conclusions

The conclusion is correct.  

References

References are correct.

Author Response

(The authors gave the same response as above.)

Reviewer 3 Report

Point 1:  Introduction section must add the main scientific contribution, methods, main advantages and limitations of the proposed antenna with respect to other antennas reported in the literature.

Point 2: The English style and grammar of all the manuscript must be significantly improved. Please revise the typing mistakes. The references placed in [] must be put at the end of the sentence, before the dot. Make sure that always there is a space between the number and measurement unit. Capitalize the first letter at the beginning of all sentences. Authors must use the same size, thick and style for all the variables and symbols used in the manuscript text, tables and figures.

Point 3: Line 108: the space is after the dot, not in front.

            Line 109: The space is after the comma, not in front.

Point 4:  The second section must add more information of the numerical simulations using ANSYS HFSS. Which were the main conditions and constraints used in the numerical models?

Point 5: Line 118: Specify the resonating wavelength or frequency for which the authors made the design.  

Point 6: Lines 153-166: The paragraph has many confusing sentences. Explain which is the impedance bandwidth required and what type of excitation did you use for the data in Table 2.

Point 7: Lines 167: In Table 2, it is easier to follow the data, by putting a “-” instead of blank space. For the first raw in Table 2, what is the value of epsilon r? In Table 2, can the authors explain how did they get the measurement results? Also, explain why a quality factor, Q is not given in the first raw (simulated case).

Point 8: Table 3 must be completed with “-“, when necessary.

Point 9: Line 238: The resolution in Figure 6 must be improved. The legend is more vivid than the representation.

Point 10: Results and Discussion section: The results section must include more detailed data about the experimental setup and this information should be provided at the beginning of the section.

Point 11: Conclusion section must be improved based on the above comments.

Point 12: This manuscript should include more recent references between 2016 and 2021. 

Author Response

(The authors gave the same response as above.)

Round 2

Reviewer 1 Report

The authors appropriately reflected the comments to improve the quality of the paper. Nevertheless, there is still room for improvement, so the paper can be accepted after the following minor revisions.

  1. I asked the authors to revise the basic grammatical elements of the manuscript, but there are still many errors. For example, capital letters should be used in line 51, and there are space typos in lines 71, 76, and 79. Please the authors double-check the revised manuscript and guarantee the basic quality of the paper.

Additionally, why do you need ',.' after the reference? This makes the sentence very confusing.

  1. To comment 6 in the first review, the authors responded that "a minimum bandwidth of 1 GHz is essential for 5G antennas in the millimeter wave frequency band." Please the authors add this description to the manuscript.
  2. For comment 8 of the first review, the authors did not provide enough explanation.

“Please add a comparison of the bandwidth and the electrical size at the center frequency in Table 6. Since the proposed antenna has a large electrical size, it seems that it is necessary to additionally compare the gain in this regard.”

Please answer this comment and add a description to the manuscript as well.

Author Response

Dear Editor,

Thank you for allowing a resubmission of our manuscript, with an opportunity to address the reviewers’ comments.

This research article has been submitted to Sensors with manuscript ID Sensors-1162560. We received a minor correction decision of the manuscript on 7th April  2021 with reviewers’ recommendations to improve this article. All the review comments have been answered for each reviewer. This article focuses on design of Dielectric Resonator Antenna at millimeter wave frequency bands for 5G applications. A systematic investigation and analysis have been done in designing the antenna. This article will be helpful for the researchers in 5G.      

     We are uploading (a) our point-by-point response to the comments (response to two reviewers in two files), (b) an updated manuscript with “Track Changes” function highlighting indicating changes, and (c) a clean updated manuscript without highlights (PDF main document).

Best regards,

Assoc. Prof. Dr. Mohd Haizal bin Jamaluddin

Wireless Communication Center

School of Electrical Engineering

Universiti Teknologi Malaysia, Johor, Malaysia

Reviewer 3 Report

Put the reference before the dot. Examples of correct introduction of references:

Line 43:  frequencies [1].

Line 47: mode [3], [4].

And so on. 

Author Response

(The authors gave the same response as above.)
